# Comparison of RNA-Sequencing Methods for Degraded RNA

**DOI:** 10.3390/ijms25116143

**Published:** 2024-06-02

**Authors:** Hiroki Ura, Yo Niida

**Affiliations:** 1Center for Clinical Genomics, Kanazawa Medical University Hospital, 1-1 Daigaku, Uchinada, Kahoku 920-0923, Japan; niida@kanazawa-med.ac.jp; 2Division of Genomic Medicine, Department of Advanced Medicine, Medical Research Institute, Kanazawa Medical University, 1-1 Daigaku, Uchinada, Kahoku 920-0923, Japan

**Keywords:** transcriptome, RNA-Seq, degraded RNA, gene expression

## Abstract

RNA sequencing (RNA-Seq) is a powerful technique and is increasingly being used in clinical research and drug development. Currently, several RNA-Seq methods have been developed. However, the relative advantage of each method for degraded RNA and low-input RNA, such as RNA samples collected in the field of clinical setting, has remained unknown. The Standard method of RNA-Seq captures mRNA by poly(A) capturing using Oligo dT beads, which is not suitable for degraded RNA. Here, we used three commercially available RNA-Seq library preparation kits (SMART-Seq, xGen Broad-range, and RamDA-Seq) using random primer instead of Oligo dT beads. To evaluate the performance of these methods, we compared the correlation, the number of detected expressing genes, and the expression levels with the Standard RNA-Seq method. Although the performance of RamDA-Seq was similar to that of Standard RNA-Seq, the performance for low-input RNA and degraded RNA has decreased. The performance of SMART-Seq was better than xGen and RamDA-Seq in low-input RNA and degraded RNA. Furthermore, the depletion of ribosomal RNA (rRNA) improved the performance of SMART-Seq and xGen due to increased expression levels. SMART-Seq with rRNA depletion has relative advantages for RNA-Seq using low-input and degraded RNA.

## 1. Introduction

Transcriptome analysis by RNA sequencing (RNA-Seq) is a powerful tool for genome-wide quantification of RNA expression with high sensitivity [1,2,3,4,5]. In recent years, many RNA-Seq methods have developed rapidly, enabling the analysis of differential expression for the transcriptome in many fields, including cancer research and clinical diagnosis [6,7,8]. However, the accuracy of RNA-Seq is highly dependent upon the quality and quantity of RNA input. Although RNA-seq has been used in various fields of clinical research, including cancer, high-quality RNA cannot be expected from tissue taken from patient organs due to contamination from dead cells, autolysis of cells, and the time required from sample collection to storage and processing [9,10,11]. The formalin-fixed paraffin-embedded (FFPE) tissue blocks enable the prolonged storage of clinical samples, preserving nucleic acids information as well as tissue morphology. However, the FFPE processing is known to result in degraded RNA and low-input RNA, which limits the gene-expression analysis using RNA-Seq [12,13,14,15,16,17].

Recently, several RNA-Seq methods have been developed. The Standard method of RNA-Seq removes the abundant ribosomal RNA (rRNA) by Poly(A) capturing of mRNA using Oligo dT beads (Figure 1) [18]. The captured mRNAs are fragmented by heat treatment, and then, the double-stranded cDNAs are synthesized. The Standard RNA-Seq using Poly(A) capturing is currently the most common method for transcriptome analysis. However, Poly(A) capturing is less suitable for degraded RNA derived from clinical samples because most captured mRNA does not remain full-length [10]. Currently, the rRNA depletion method overcomes the limitation of Poly(A) capturing, and its performance is better for degraded RNA than that of Poly(A) capturing [11,19,20]. The rRNA depletion method is not very suitable for low-input RNA because it requires at least 100 ng levels of total RNA input [21,22].

During recent years, RNA-Seqs methods tailored for degraded RNA samples have been developed, including commercially available SMART-Seq, xGen Broad-range, and RamDA-Seq [23,24]. In SMART-Seq, the first-strand cDNAs are synthesized using the N6 primer, which is a random primer (Figure 2A). SMART-Seq takes advantage of template switching using MMLV (Moloney murine leukemia virus) reverse transcriptase to generate full-length double-stranded cDNAs from low-input RNA. In addition, the mRNA-derived libraries are enriched by the cleavage of the rRNA-derived libraries using ZapR and R-probes. In xGen, the first-strand cDNAs are synthesized using random primers (Figure 2B). xGen takes advantage of Adaptase technology, which simultaneously performs tailing and ligation to incorporate a single-stranded DNA adaptor for each first-stranded cDNA generated by Randam primers to create double-stranded cDNA, thereby covering full-length mRNA. In RamDA-Seq, the first-stranded cDNAs are synthesized using not-so-random (NSR) primers and Oligo dT primers (Figure 2C). The RNA–DNA hybrid is randomly cleaved, and then, the RamDA-Seq takes advantage of strand displacement amplification using the T4 gene 32 protein to generate full-length double-stranded cDNAs.

Several studies have compared the performance between RNA-Seq methods using Poly(A) capturing and rRNA depletion for degraded [11] and low-input RNA samples [25], including FFPE samples [25,26,27]. However, the performance evaluation of SMART-Seq, xGen Broad-range, and RamDA-Seq for degraded and low-input RNAs has not been reported. Although the rRNA depletion method has the limitation of RNA amount, SMART-Seq, xGen Broad-range, and RamDA-Seq have the possibility to be suitable not only for degraded RNA but also for low-input RNA, unlike the rRNA depletion method. It is difficult to obtain undegraded RNA from FFPE samples because the RNA is already degraded. In this study, we used the artificial degraded RNA extracted from human induced pluripotent stem cells (hiPSC), which reproducibly express many kinds of transcripts, and compared it with the original undegraded RNA [28]. This allowed for an accurate determination of the RNA degradation effects in RNA-Seq. We evaluated the performance of three commercially available RNA-Seq library preparation kits (SMART-Seq, xGen, and RamDA-Seq) compared to Standard RNA-Seq to determine the best method for transcriptome analysis using the artificially degraded RNA.

## 2. Results

### 2.1. Comparison between Standard, SMART-Seq, xGen, and RamDA-Seq for Expression Analysis

To evaluate the performance of SMART-Seq, xGen, and RamDA-Seq for expression analysis, we compared the accuracy of the gene-expression profile with Standard RNA-Seq. The correlation between biological duplicates (Standard first and second) was shown to be of high value (R = 0.976) (Figure 3A). Also, another KAPA mRNA-capturing Standard RNA-Seq (Standard KAPA) showed a similarly high value (R = 0.961). The correlation of each SMART-Seq and xGen was slightly lower than that of Standard RNA-Seq (R = 0.833 and 0.878). On the other hand, the correlation of RamDA-Seq showed a high value, similar to Standard RNA-Seq. The number of differential expression genes (DEGs) in Standard KAPA and RamDA-Seq was higher than in SMART-Seq and xGen (Figure 3B). The number of the expressing genes in Standard KAPA, SMART-Seq, and RamDA-Seq was similar to that in Standard RNA-Seq (Figure 3C). The number of the expressing genes in xGen was slightly lower than in other methods. The overall expression levels in SMART-Seq and xGen were lower overall than in Standard RNA-Seq and RamDA-Seq (Figure 3D). The expression level in RamDA-Seq was similar to that in the Standard RNA-Seq. Moreover, the ratio of the higher expressing genes (TPM > 50) was lower in SMART-Seq and xGen than in Standard RNA-Seq and RamDA-Seq (Figure 3E). The ratio of the higher expressing genes in RamDA-Seq was similar to that in the Standard RNA-Seq. The ratio of up-regulated DEGs in SMART-Seq and xGen was higher than that in Standard RNA-Seq and RamDA-Seq (Figure 3A,F). The ratio of the up-regulated DEGs in RamDA-Seq was similar to that in the Standard RNA-Seq. The majority of up-regulated DEGs in SMART-Seq and xGen were non-coding RNAs without poly-A tails (Figure 3G). On the other hand, the majority of up-regulated DEGs in Standard RNA-Seq and RamDA-Seq and down-regulated genes were protein-coding genes. These results indicated that the performance of RamDA-Seq in expression analysis was similar to that of Standard RNA-Seq. Although SMART-Seq and xGen can detect not only protein-coding RNAs (mRNA) but also non-coding RNA, the expression levels in SMART-Seq and xGen were lower than Standard RNA-Seq and RamDA-Seq.

### 2.2. Comparison between Standard, SMART-Seq, xGen, and RamDA-Seq for Low-Input RNA

To evaluate the performance of SMART-Seq, xGen, and RamDA-Seq for low-input RNA, we compared their accuracy in the gene-expression profiling with Standard RNA-Seq. The correlation of SMART-Seq 1 ng and xGen 100 ng was slightly lower than RamDA-Seq 1 ng (Figure 4A). The correlation of SMART-Seq 1 ng was higher than that of xGen 1 ng. Moreover, the correlation of SMART-Seq 10 pg was higher than RamDA-Seq 10 pg. The number of the detected expressing genes in SMART-Seq 1 ng was similar to that in RamDA-Seq 1 ng, and the number was higher than not only xGen 1 ng but also xGen 100 ng (Figure 4B). The number of the detected expressing genes in SMART-Seq 10 pg was higher than in RamDA-Seq 10 pg. The overall expression level in RamDA-Seq 1 ng was higher overall than SMART-Seq 1 ng, xGen 100 ng, and xGen 1 ng (Figure 4C). However, the expression level in RamDA-Seq 10 pg was significantly decreased. On the other hand, the expression level in SMART-Seq 10 pg was not significantly decreased. Each correlation of SMART-Seq was higher than that of xGen and RamDA-Seq (Figure 4D). The ratio of down-regulated DEGs in SMART-Seq 10 pg was slightly higher than that in xGen 1 ng and RamDA-Seq 10 pg (Figure 4E). The respective down-regulated genes in SMART-Seq 10 pg, xGen 1 ng, and RamDA-Seq 10 pg were almost not detected in SMART-Seq 10 pg, xGen 1 ng, and RamDA-Seq 10 pg (Figure 4F). SMART-Seq is better than xGen and RamDA-Seq at RNA 10 pg, although RamDA-Seq is better than SMART-Seq at RNA 1 ng, indicating that SMART-Seq is better for low-input RNA.

### 2.3. Comparison between Standard, SMART-Seq, xGen, and RamDA-Seq for Degraded RNA

To evaluate the performance of SMART-Seq, xGen, and RamDA-Seq for degraded RNA such as RNA from the formalin-fixed paraffin-embedded (FFPE) samples, we compared the accuracy of their gene-expression profiles with Standard RNA-Seq. First, we generated the artificially degraded RNA by heat treatment (Figure 5A). Two different degraded RNA samples (RIN 4 and RIN 1 (RIN:RNA integrity number)) were made at the different heat-treatment times. The correlation of SMART-Seq was also high in the degraded RIN1 RNA sample (Figure 5B). On the other hand, the correlation of the RamDA-Seq RNA sample was decreased in the degraded RIN1, although the correlation was high in the degraded RIN 4 RNA sample. In xGen, the correlation was already decreased for the RIN 4 RNA sample, whereas the correlation for the RIN 1 RNA sample was similar to that for the RIN4 RNA sample. The number of the detected expressing genes by SMART-Seq in RIN4 and RamDA-Seq in RIN 4 was higher than that of xGen in RIN4 (Figure 5C). The number of detected expressing genes by SMART-Seq in RIN1 was slightly decreased, but the number of detected genes by RamDA-Seq in RIN 1 was decreased significantly. In xGen, the number of detected genes in RIN1 was similar to that in RIN 4. Next, we calculated the correlation in each method (Figure 5D). The correlation in RIN 4 and RIN 1 for SMART-Seq and xGen was similar to the correlations for each undegraded RNA sample for SMART-Seq and xGen. On the other hand, for RamDA-Seq, although the correlation in RIN4 was not decreased, the correlation in RIN1 was significantly decreased. The expression level in RIN 4 for SMART-Seq and RamDA-Seq was similar to Standard RNA-Seq (Figure 5E). The expression level in RIN 1 for RamDA-Seq was significantly decreased. However, the expression level in RIN 1 for SMART-Seq was not significantly decreased. In xGen, the expression levels of both RIN 4 and RIN1 were similarly decreased. The coverage in SMART-Seq and xGen tended to be biased towards the 5′ end of the transcripts, whereas the coverage in RamDA-Seq was uniform within the transcripts (Figure 5F). However, the coverage in RamDA-Seq tended to be biased towards the 3′ end of the transcripts in RIN 4 and RIN1 for RamDA-Seq, depending on the extent of RNA degradation. On the other hand, the coverage in SMART-Seq was similar for the RNA degradation samples. The coverage in xGen was not uniform, even for undegraded RNA samples. These results indicated that the performance of SMART-Seq is better than xGen and RamDA-Seq for a degraded RNA sample, although RamDA-Seq is still useful on less-degraded RNA samples.

### 2.4. Improvement of SMART-Seq and xGen by Ribosomal RNA Depletion

We investigated the ratio of ribosomal RNA (rRNA) because of the possibility that the expression of rRNA affects the expression levels in SMART-Seq and xGen (Figure 6A). The ratio of rRNA in Standard RNA-Seq and RamDA-Seq was low, whereas the ratio of rRNA in SMART-Seq was approximately 30%. Moreover, the ratio of rRNA was about 80% in xGen. To evaluate the performance of SMART-Seq and xGen with rRNA depletion and/or mRNA capture, we compared the accuracy of the gene-expression profile with Standard RNA-Seq. The ratio of rRNA in SMART-Seq and xGen with rRNA depletion and/or mRNA capture was very low (Figure 6B). The correlation in SMART-Seq with rRNA depletion was increased at 100 ng RNA. However, the correlation was decreased at 10 pg RNA (Figure 6C). In xGen with rRNA depletion and mRNA capture, the correlation was increased. The number of the detected expression genes in SMART-Seq with rRNA depletion at 100 ng RNA and xGen with rRNA depletion and mRNA capture remained largely unchanged (Figure 6D). In SMART-Seq with rRNA depletion at 10 pg RNA, only a few were detected. The expression levels in SMART-Seq with rRNA depletion at 100 ng RNA and xGen with rRNA depletion and mRNA capture were increased, whereas the expression levels were decreased in SMART-Seq with rRNA depletion at 10 pg RNA (Figure 6E). The coverage in SMART-Seq with rRNA depletion at 10 pg RNA was slightly less uniform than at 100 ng (Figure 6F). The coverage in xGen with rRNA depletion and mRNA capture was more uniform than that in xGen without rRNA depletion and mRNA capture (Figure 5F and Figure 6F). These results indicated that rRNA depletion improved the performance of SMART-Seq and xGen. However, a certain amount of RNA was required for rRNA depletion.

### 2.5. Improvement of SMART-Seq and xGen for Degraded RNA by rRNA Depletion

To evaluate the performance of SMART-Seq and xGen with rRNA depletion for degraded RNA, we compared their accuracy regarding the gene-expression profile with Standard RNA-Seq. Similarly, the ratio of rRNA for degraded RNA was low in RamDA-Seq, whereas the ratio of rRNA was approximately 30% in SMART-Seq and about 90% in xGen (Figure 7A). The ratio of rRNA in SMART-Seq with rRNA depletion for degraded RNA was very low (Figure 7B). In xGen with rRNA depletion, the ratio of rRNA in RIN 4 was low, while the ratio in RIN 1 was still around 20%. In SMART-Seq with rRNA depletion for degraded RNA, the correlation for RIN 4 was slightly increased but not for RIN 1 (Figure 7C). On the other hand, in xGen with rRNA depletion, the correlation was relatively increased for RIN 4 but decreased for RIN 1. The number of detected expressing genes in SMART-Seq with rRNA depletion remained largely unchanged (Figure 7D). In xGen, the detected expressing gene number for RIN 4 was increased, but the detected number for RIN 1 was decreased. The expression levels for RIN 4 and RIN 1 in SMART-Seq were slightly increased and were similar to those in the Standard RNA-Seq (Figure 7E). Although the expression levels for RIN 4 were significantly increased, the expression levels for RIN 1 were not increased in xGen. The coverage of both RIN 4 and RIN 1 in SMART-Seq was uniform (Figure 7F). On the other hand, the coverage of RIN 4 was uniform in xGen, but the coverage of RIN 1 was still not uniform. These results indicated that rRNA depletion slightly improved the performance of SMART-Seq for degraded RNA. In xGen, rRNA depletion significantly improved the performance of xGen for moderately degraded RNA. However, rRNA depletion did not improve the performance of xGen for highly degraded RNA.

## 3. Discussion

Transcriptome analysis by RNA-Seq is a powerful tool for the quantification of whole genome transcripts obtained from various genes in human disease and developmental studies. However, it is difficult to extract high-quality RNA from clinical specimens due to contamination from dead cells, autolysis of cells, and the time required from sample collection to storage and processing. Clinical biospecimens, such as cancer tissues, are typically stored as FFPE blocks, which are a valuable source of material for biomedical research. FFPE blocks enable the prolonged storage of clinical samples, preserving nucleic acids information as well as tissue morphology. However, FFPE processing and tissue storage are known to affect RNA quality, which limits the Standard RNA-Seq method, including mRNA capture using Oligo dT. For degraded RNA, it is required that the RNA-Seq method does not use mRNA capture using Oligo dT. With the development and advancement of RNA-Seq methods, many library preparation methods have become available. Here, we used the artificially degraded RNA extracted from hiPSCs, which reproducibly express many kinds of transcripts. Then, we compared it to the original undegraded RNA and evaluated the performance of three commercially available RNA-Seq library preparation kits (SMART-Seq, xGen, and RamDA-Seq) compared to Standard RNA-Seq to determine the best method for transcriptome analysis using the artificially degraded RNA, such as the RNA extracted from an FFPE sample.

In undegraded RNA, the number of detected genes was almost the same between Standard RNA-Seq, SMART-Seq, and RamDA-Seq. The number of detected genes in xGen was fewer than others. The majority of up-regulated differential expression genes (DEGs) in SMART-Seq and xGen were non-coding RNAs without poly-A tails. Standard RNA-Seq cannot detect non-coding genes due to mRNA capturing using Oligo dT, suggesting that SMART-Seq and xGen have an advantage in analyzing the expression of both mRNA and non-coding genes. The expression pattern and expression level of RamDA-Seq were more similar to that of Standard RNA-Seq than SMART-Seq or xGen. These results suggest that RamDA-Seq is better than SMART-Seq and xGen for undegraded RNA. The correlation and the number of detected genes in SMART-Seq for low input are better than xGen and RamDA-Seq, suggesting that the performance of SMART-Seq for low-input RNA was better than others.

For degraded RNA, the performance of SMART-Seq was shown to be relatively high regardless of the degree of RNA degradation. On the other hand, the performance of xGen was shown to be low at any degree of RNA degradation. The performance of RamDA-Seq was shown to be relatively low for highly degraded RNA. Furthermore, the coverage in RamDA-Seq tended to be biased towards the 3′ end of transcripts, depending upon the degree of RNA degradation, suggesting that RamDA-Seq is not suitable for degraded RNA. These results suggest that the performance of SMART-Seq for degraded RNA was higher than others. In xGen, many rRNA-derived transcripts were expressed. The rRNA-derived transcripts were expressed in SMART-Seq, although not as abundantly as in xGen. Since it is possible that rRNA-derived transcripts affect expression levels, we evaluated the performance of SMART-Seq and xGen when they are depleted of rRNA-derived transcripts. As expected, rRNA depletion improved the performance of SMART-Seq and xGen. However, the performance of SMART-Seq with rRNA depletion has decreased for low-input RNA, suggesting that it is required, for a certain amount of RNA, to perform rRNA depletion. The performance of SMART-Seq with rRNA depletion for degraded RNA was also improved by rRNA depletion. On the other hand, the performance of xGen for highly degraded RNA was not improved, although the performance for moderately degraded RNA was improved. These results suggest that SMART-Seq has an advantage for degraded RNA.

In this study, we compared the performance of three different methods using artificially degraded RNA samples, such as FFPE samples. In summary, SMART-Seq is a better RNA-Seq method for degraded RNA and low-input RNA than xGen and RamDA-Seq. However, actual RNA extracted from an FFPE sample often contains artifact alterations caused by formalin fixation as well as RNA degradation [29]. In the future, we will plan to evaluate the performance of SMART-Seq for FFPE samples.

## 4. Materials and Methods

### 4.1. Total RNA Extraction and RNA Degradation by Heat-Treatment

Total RNA from human induced pluripotent stem cells (iPSCs) was extracted with TRIzol reagent (Thermo Fisher Scientific, Waltham, MA, USA) according to the manufacturer’s instructions [30]. The RNA integrity number (RIN) indicates the degree of RNA degradation. The RIN 4 RNAs were obtained from the total RNA by heat treatment at 94 °C for 1 min, and RIN 1 RNAs were obtained by heat treatment at 94 °C for 5 min. The RNA concentration and purity were measured spectrophotometrically (Nanodrop); RIN was measured by TapeStaion 4200 with RNA Screen Tape (Agilent Technologies, Santa Clara, CA, USA).

### 4.2. Standard RNA-Seq Library Preparation

The mRNA from the total RNA was captured using a NEBNext Poly(A) mRNA Magnetic Isolation Module (New England Biolabs, Ipswich, MA, USA) or KAPA mRNA capture kit (Kapa Biosystems, Inc., Wilmington, MA, USA), and then, Standard RNA-Seq libraries were synthesized from the captured mRNA using NEBNext Ultra II directional RNA Library Prep Kit for Illumina according to the manufacturer’s instructions (Figure 1). The quality of the libraries was assessed using the TapeStation 4200 with the High Sensitivity D100 Screen Tape.

### 4.3. SMART-Seq Library Preparation

SMART-Seq libraries were synthesized from total RNA using a SMART-Seq Stranded Kit (TaKaRa Bio USA, Mountain View, CA, USA) according to the manufacturer’s instructions (Figure 2A). The first-strand cDNAs were synthesized from the total RNA using an N6 random primer, and then, the double-strand cDNAs were synthesized by template switching. After the addition of Illumina adapters, the double-strand cDNAs derived from rRNA were cleavaged by scZapR and scR-Probes. The uncleavaged libraries were amplified over 12 cycles. The quality of the libraries was assessed using the TapeStation 4200 with High Sensitivity D100 Screen Tape.

### 4.4. xGen Library Preparation

xGen libraries were synthesized from the total RNA using IDT xGen Broad-Range RNA Library Prep Kit (Integrated DNA Technologies, Coralville, LA, USA) according to the manufacturer’s instructions (Figure 2B). The first-strand cDNAs were synthesized from the heat-treated total RNA using a random primer and were added to the adapter by Adaptase technology. The double-strand cDNAs were extended and then ligased to the adapter. The libraries from the adapter-ligated double-strand cDNAs were amplified by indexing PCR. In rRNA depletion and poly(A) capturing, 100 ng of RNA were used for xGen library preparation. The quality of the libraries was assessed using the TapeStation 4200 with High Sensitivity D100 Screen Tape.

### 4.5. RamDA-Seq Library Preparation

RamDA-Seq libraries were synthesized from total RNA using a GenNext RamDA-seq Single Cell Kit (New England Biolabs, Ipswich, MA, USA) according to the manufacturer’s instructions (Figure 2C). The first-strand cDNAs were synthesized from total RNA using a not-so-random (NSR) primer and Oligo-dT primer. The RNA–DNA hybrid was cleavaged by DNase I treatment. The strands were displaced by amplification using the T4 gene 32 protein, and then, the second-strand cDNAs were synthesized from displacement strands using an NSR primer. The libraries were prepared from the second-strand cDNAs using the Nextera XT DNA library Preparation Kit (Illumina, San Diego, CA, USA) for Illumina sequencing, as described previously [31]. The quality of the libraries was assessed using the TapeStation 4200 with the High Sensitivity D100 Screen Tape.

### 4.6. rRNA Depletion

The rRNA-depleted RNA was obtained using a NEBNext rRNA Deletion Kit v2 (New England Biolabs, Ipswich, MA, USA) according to the manufacturer’s instructions. The rRNAs were hybridized specifically to single-stranded DNA probes. The rRNA-depleted RNAs were obtained by degradation of the hybrid DNA–rRNA using RNase H Enzyme.

### 4.7. Sequencing and Generation of FASTQ Files

The RNA-Seq libraries were quantified using an HS Qubit dsDNA assay (Thermo Fisher Scientific, Waltham, MA, USA) and a GenNext NGS Library Quantification Kit (New England Biolabs, Ipswich, MA, USA). The RNA-Seq libraries were sequenced on the Illumina NextSeq (2 × 75 bp) following the standard Illumina protocol (Illumina, San Diego, CA, USA). The FASTQ files containing the molecular barcodes were generated using bcl2fastq software (v2.19.1) (Illumina).

### 4.8. Data Analysis

The FASTQ files were aligned to the reference human genome (hg38) using HISAT2 (version 2.1.0) [32]. Subsequently, the StringTie algorithm (version v1.3.4d) [33] was used with default parameter settings to assemble RNA-Seq alignments into annotated transcripts and estimate their respective expression level. The expression of these transcripts was normalized using the transcripts per million (TPM) algorithm. For differential expression analysis, we used R packages TCC [34]. For analysis and interpretation, we used the following R packages and analysis approach described previously [31,35,36].

## 5. Conclusions

In summary, the performance in low-input RNA and degraded RNA has decreased, although the performance of RamDA-Seq was similar to Standard RNA-Seq. The performance of SMART-Seq was better than xGen and RamDA-Seq in low-input RNA and degraded RNA. Furthermore, rRNA depletion improved the performance of SMART-Seq and xGen due to increased expression levels. SMART-Seq with rRNA depletion has relative advantages for RNA-Seq using low-input and degraded RNA.

## Figures and Tables

**Figure 1 ijms-25-06143-f001:**
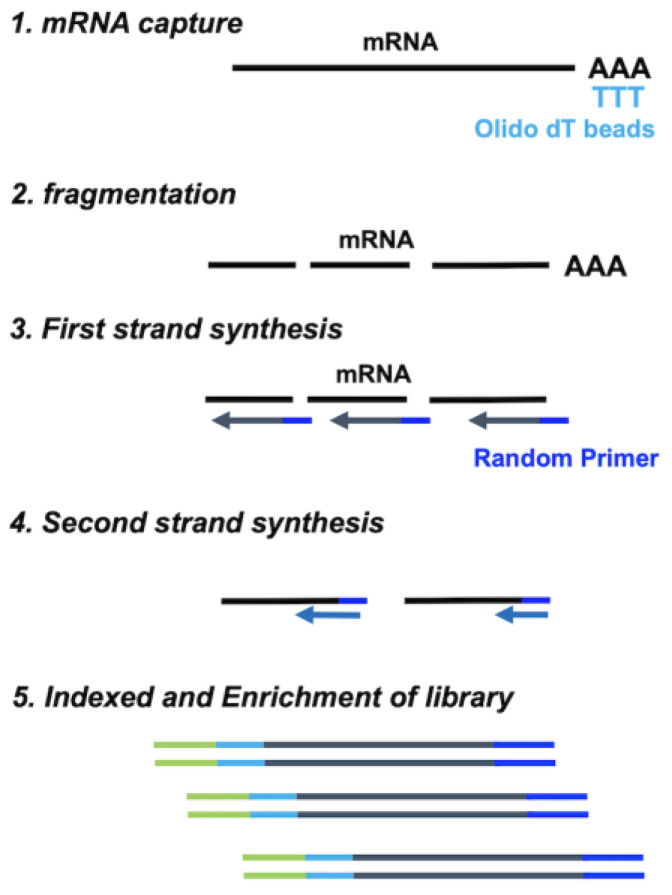
Library preparation workflow of Standard RNA-Seq using Oligo dT beads.

**Figure 2 ijms-25-06143-f002:**
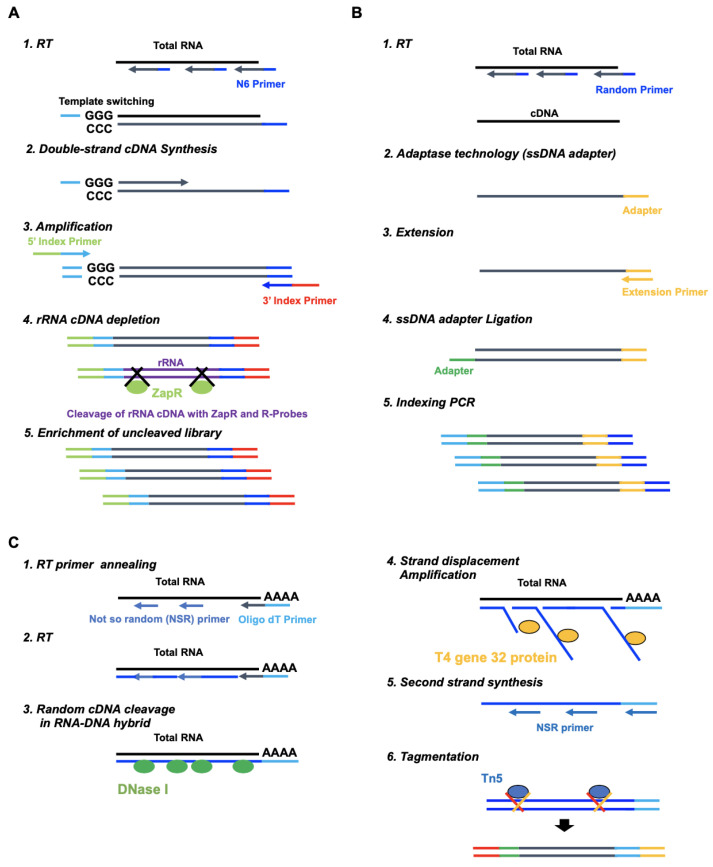
Library preparation workflow. (**A**) SMART-Seq, (**B**) xGen, and (**C**) RamDA-Seq.

**Figure 3 ijms-25-06143-f003:**
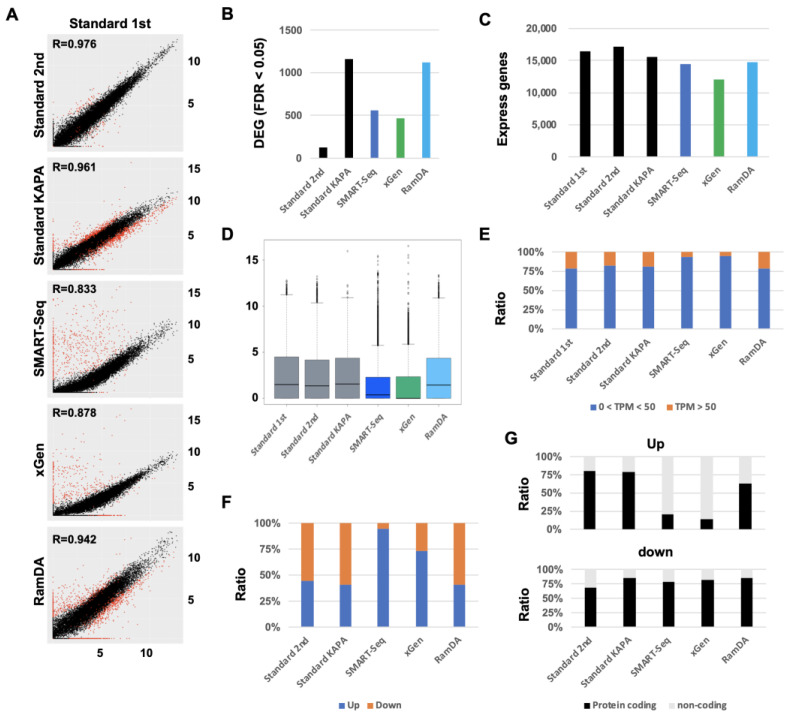
Comparison of the performance of SMART-Seq, xGen, and RamDA-Seq for undegraded RNA. (**A**) Scatter plot [log2 (TPM + 1)] of total genes. Red spots indicate the differential expression genes (DEGs) (FDR < 0.05). (**B**) The number of DEGs. (**C**) The number of detected expressing genes. (**D**) Boxplot of total genes [log2 (TPM + 1)]. (**E**) The ratio of genes with different expression levels (0 < TPM < 50 and TPM > 50). (**F**) The ratio of up-regulated genes and down-regulated genes. (**G**) The ratio of protein-coding genes and non-coding genes. Upper: upregulated genes. Down: down-regulated genes.

**Figure 4 ijms-25-06143-f004:**
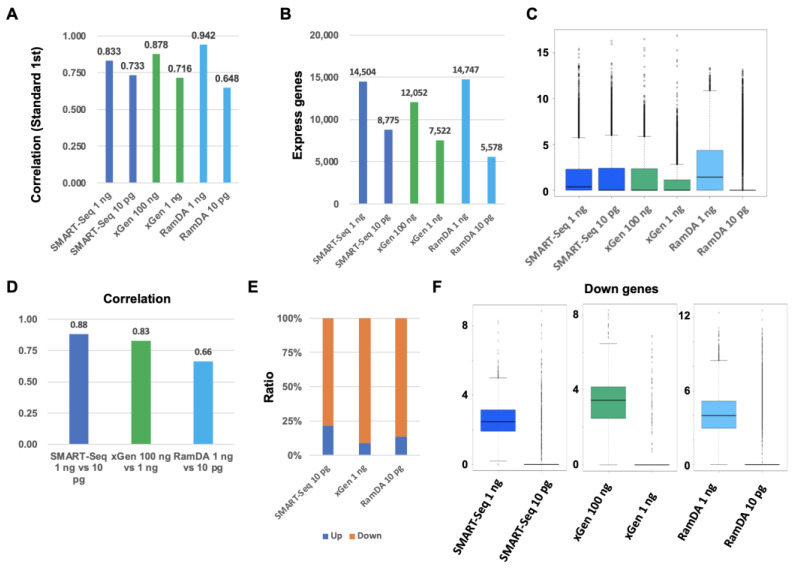
Comparison of the performance of SMART-Seq, xGen, and RamDA-Seq for low-input RNA. (**A**) Each correlation with Standard RNA-Seq [log2 (TPM + 1)]. (**B**) The number of detected expressing genes. (**C**) Boxplot of expressing genes [log2 (TPM + 1)]. (**D**) Each correlation with each method for normal input RNA [log2 (TPM + 1)]. (**E**) The ratio of up-regulated genes and down-regulated genes. (**F**) Boxplot of each down-regulated gene [log2 (TPM + 1)].

**Figure 5 ijms-25-06143-f005:**
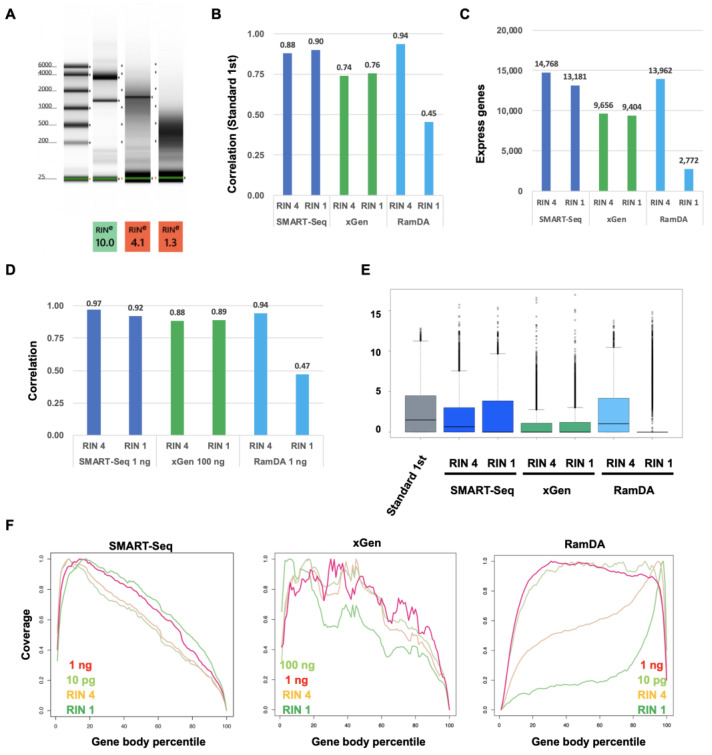
Comparison of the performance of SMART-Seq, xGen, and RamDA-Seq for degraded RNA. (**A**) The electrophoresis of RNA sample. (RNA integrity number (RIN)) (**B**) Each correlation with Standard RNA-Seq [log2 (TPM + 1)]. (**C**) The number of detected expressing genes. (**D**) Each correlation with each method for normal input RNA [log2 (TPM + 1)]. (**E**) Boxplot of total genes [log2 (TPM + 1)]. (**F**) The distribution of mapped reads on gene bodies.

**Figure 6 ijms-25-06143-f006:**
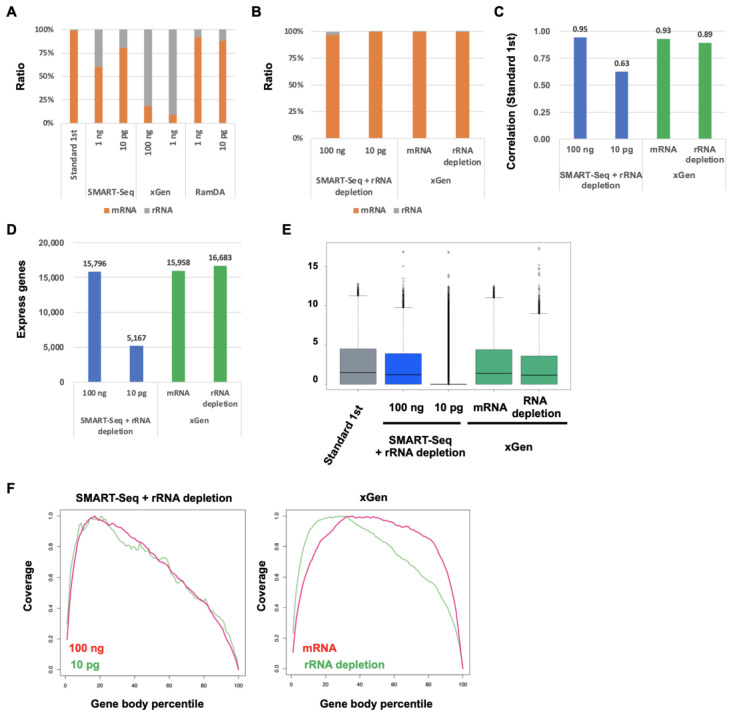
Comparison of the performance of SMART-Seq, xGen, and RamDA-Seq with rRNA depletion (**A**,**B**) The ratio of mRNA and rRNA. (**C**) Each correlation with Standard RNA-Seq [log2 (TPM + 1)]. (**D**) The number of detected expressing genes. (**E**) Boxplot of total genes [log2 (TPM + 1)]. (**F**) The distribution of mapped reads on gene bodies.

**Figure 7 ijms-25-06143-f007:**
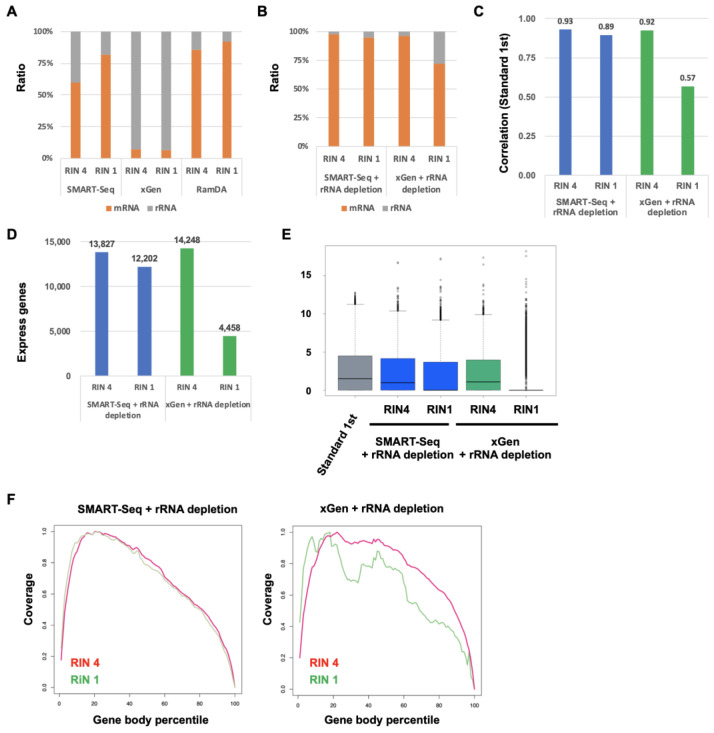
Comparison of the performance of SMART-Seq, xGen, and RamDA-Seq with rRNA depletion for degraded RNA (**A**,**B**) The ratio of mRNA and rRNA. (**C**) Each correlation with Standard RNA-Seq [log2 (TPM + 1)]. (**D**) The number of detected expressing genes. (**E**) Boxplot of total genes [log2 (TPM + 1)]. (**F**) The distribution of mapped reads on gene bodies.

## Data Availability

The raw data have been deposited in the Sequence Read Archive database of NCBI (GSE266382).

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
