# Peer review of "Comparison of RNA-Sequencing Methods for Degraded RNA"

_ijms, 2024, doi:10.3390/ijms25116143_

Round 1

Reviewer 1 Report

Comments and Suggestions for Authors

The paper by Ura and Niida seeks to evaluate the performance of commercially available RNA-Seq library preparation kits that use random primers compared to what the authors characterize as the Standard RNA-Seq method.

At first, it should be clear and illustrated what is the standard RNA-Seq method. In this sense, the authors begin by discussing ribosomal RNA depletion methods. They focus on Poly(A) capturing of mRNA using Oligo dT beads and refer to this method as the standard, which, in my opinion, does not accurately describe the reality. In fact, among the several techniques for rRNA removal in RNA-seq, ribosomal RNA depletion kits are particularly popular and widely used in the research community. This background should be described in more depth in the text.

Furthermore, it is not explicit what the authors are comparing since the RNA-seq library preparation kits tested and the rRNA depletion kit as the Standard RNA-seq method serve different, yet complementary, purposes in the workflow of RNA sequencing.

Author Response

Dear Reviewer,

Thank you for giving us the opportunity to submit a revised draft of our manuscript titled “Comparison of RNA sequencing methods for degraded RNA” to the International Journal of Molecular Sciences. We appreciate the time and effort that you have dedicated to providing your valuable feedback on our manuscript. We are grateful to you for your insightful comments on our paper. We have been able to incorporate changes to reflect all of your suggestions. We highlighted the changes within the manuscript.

We have revised to emphasize this point and added Figure 1 (Line 42-43 in Page 1 and Line 47-50, 70-73 in Page 2). The ribosomal RNA depletion kits are widely used for degraded RNA and several studies compare the performance. However, it is not reported that the performance evaluation of commercially available SMART-Seq, xGen Broad-range and RamDA-Seq. 

Sincerely,

Hiroki Ura, Ph.D.

Center for Clinical Genomics

Kanazawa Medical University Hospital

  • Daigaku, Uchinada, Kahoku, Ishikawa, 920-0293, JAPAN

Phone No: +81 076-286-2211

Email Address: h-ura@kanazawa-med.ac.jp

Reviewer 2 Report

Comments and Suggestions for Authors

In this manuscript, Ura and Niida reported a systematic comparison of RNA sequencing methods, like SMART-Seq, xGen and RamDA-Seq. They compared the performance of these methods with Standard-Seq using degraded or undegraded RNAs. This topic is of high importance and interest for people working on the RNA seq. At last, they found SMART-Seq is the best for degraded RNA and low input RNA. The whole project is well designed, and the results support the final conclusion. However, the English needs to be polished. Some language mistakes are observed.

1.       Why RamDA was not evaluated in the condition of rRNA depletion or mRNA purification?

2.       As the authors try to present an unbiased study of different sequence method, it is better to keep the same condition for comparison. In the rRNA depletion and mRNA purification condition, rRNA was used for SMART-seq in 300 ng and 10 pg, while there was not information about the RNA quantity for xGen. This must be added.

3.       One grammar mistake: line 87, The expression level in RamDA-Seq was similar to in the Standard RNA-Seq. “that” should be added in front of “in”. This same mistake for line 93, 115, 123,

4.       Figure 2D has some items in Japanese. Please correct it.

5.       Line 140: sample>>samples

6.       Line 141:difference>>different

7.       Figure 4F: unify the color usage for the same item

8.       Line 184: express>>expression

9.       I do not understand the sentence in line 190: The coverage in SMART-Seq with rRNA depletion at 300 ng RNA and xGen with rRNA depletion and mRNA capture was more uniform

10.   Line 266:Iit>>it

11.   Line 276:SMAR-Seq>>SMART-Seq

Comments on the Quality of English Language

1.       One grammar mistake: line 87, The expression level in RamDA-Seq was similar to in the Standard RNA-Seq. “that” should be added in front of “in”. This same mistake for line 93, 115, 123,

2.       Figure 2D has some items in Japanese. Please correct it.

3.       Line 140: sample>>samples

4.       Line 141:difference>>different

5.       Figure 4F: unify the color usage for the same item

6.       Line 184: express>>expression

7.       I do not understand the sentence in line 190: The coverage in SMART-Seq with rRNA depletion at 300 ng RNA and xGen with rRNA depletion and mRNA capture was more uniform

8.       Line 266:Iit>>it

9.       Line 276:SMAR-Seq>>SMART-Seq

Author Response

Dear Reviewer,

Thank you for giving us the opportunity to submit a revised draft of our manuscript titled “Comparison of RNA sequencing methods for degraded RNA” to the International Journal of Molecular Sciences. We appreciate the time and effort that you have dedicated to providing your valuable feedback on our manuscript. We are grateful to you for your insightful comments on our paper. We have been able to incorporate changes to reflect all of your suggestions. We highlighted the changes within the manuscript.

Here is a point-by point response to your comments and concerns.

Point 1: Why RamDA was not evaluated in the condition of rRNA depletion or mRNA purification?

Response 1: It is expected that RamDA-Seq do not improved in the condition rRNA depletion and mRNA purification because the coverage of RamDA-Seq for degraded RNA were already affected.

Point 2: As the authors try to present an unbiased study of different sequence method, it is better to keep the same condition for comparison. In the rRNA depletion and mRNA purification condition, rRNA was used for SMART-seq in 300 ng and 10 pg, while there was not information about the RNA quantity for xGen. This must be added.

Response 2: We have corrected and revised this point (Line 355-356 in Page 11 and Figure 6).

Point 3:  One grammar mistake: line 87, The expression level in RamDA-Seq was similar to in the Standard RNA-Seq. “that” should be added in front of “in”. This same mistake for line 93, 115, 123,

Response 3: We have corrected (Line 102 and 106 in Page 4 and Line 131 and 140 in Page 5).

Point 4: Figure 2D has some items in Japanese. Please correct it.

Response 4: We have corrected in Figure 2D.

Point 5:  Line 140: sample>>samples

Response 5: We have corrected (Line 160 in Page 5).

Point 6:  Line 141:difference>>different

Response 6: We have corrected (Line 161 in Page 5).

Point 7:  Figure 4F: unify the color usage for the same item

Response 7: We have corrected in Figure 5F.

Point 8: Line 184: express>>expression

Response 8: We have corrected (Line 206 in Page 7).

Point 9:  I do not understand the sentence in line 190: The coverage in SMART-Seq with rRNA depletion at 300 ng RNA and xGen with rRNA depletion and mRNA capture was more uniform

Response 9: We have revised to emphasize this point (Line 212-215 in Page 8).

Point 10: Line 266:Iit>>it

Response 10: We have corrected (Line 300 in Page 10).

Point 11:   Line 276:SMAR-Seq>>SMART-Seq

Response 11: We have corrected (Line 310 in Page 10).

Sincerely,

Hiroki Ura, Ph.D.

Center for Clinical Genomics

Kanazawa Medical University Hospital

  • Daigaku, Uchinada, Kahoku, Ishikawa, 920-0293, JAPAN

Phone No: +81 076-286-2211

Email Address: h-ura@kanazawa-med.ac.jp

Round 2

Reviewer 1 Report

Comments and Suggestions for Authors

I want to acknowledge and thank the authors for the revisions made, which have improved the manuscript's clarity. However, the fact remains that the study's aims and design approach are skewed and lack comprehensiveness. The SMART-Seq, xGen Broad-range, and RamDA-Seq kits are library preparation kits used in RNA sequencing workflows, but they are not rRNA depletion kits themselves. Therefore, it's not appropriate to compare them with a specific rRNA depletion method, that is not actually the standard.

Author Response

Dear Reviewer,

We appreciate the time and effort that you have dedicated to providing your valuable feedback on our manuscript. We are grateful to you for your insightful comments on our paper. We have been able to incorporate changes to reflect all of your suggestions. We highlighted the changes within the manuscript.

We have revised to emphasize this point and added Figure 1 (Line 74-76 in Page 2). Several studies compared the performance between Poly(A) capturing and rRNA depletion. However, it is still not reported that the performance evaluation of commercially available SMART-Seq, xGen Broad-range and RamDA-Seq compare to Poly(A) capturing.   

Sincerely,

Hiroki Ura, Ph.D.

Center for Clinical Genomics

Kanazawa Medical University Hospital

  • Daigaku, Uchinada, Kahoku, Ishikawa, 920-0293, JAPAN

Phone No: +81 076-286-2211

Email Address: h-ura@kanazawa-med.ac.jp
